# Targeted Muscle Reinnervation (TMR) or Regenerative Peripheral Nerve Interface (RPNI) for pain prevention in patients with limb amputation: A protocol for a systematic review and meta-analysis

**Jesús del Moral Preciado**[1], **David Gurpegui**[1]*, **Montserrat Royo**[2], **Bernardo Hontanilla**[1]

**1** Departamento de Cirugía Plástica, Estética y Reparadora, Clínica Universidad de Navarra, Pamplona, España, **2** Biblioteca de Ciencias, Universidad de Navarra, Pamplona, España

* dgurpegui@unav.es

## Abstract

### Introduction

Regenerative Peripheral Nerve Interface (RPNI) and Targeted Muscle Reinnervation (TMR) have demonstrated superior outcomes compared to classical amputation in prophylactic prevention of pain, primarily by reducing the incidence of symptomatic neuromas, residual limb pain, and phantom limb pain. However, direct comparisons between these two techniques remain limited. Furthermore, their comparative effectiveness across diverse patient demographics (including age, sex, and comorbidities) and surgical variables (amputation level, etiology, and nerve handling) has not been systematically evaluated. Therefore, the objective of this systematic review and meta-analysis is to synthesize the available evidence to determine the comparative safety and efficacy of primary TMR or RPNI.

### Methods and design

This review will be conducted following the methodological guidance of the *Cochrane Handbook for Systematic Reviews of Interventions*. A comprehensive electronic search will be performed in the Cochrane Central Register of Controlled Trials (CENTRAL), Web of Science, Scopus, PubMed, and MedRxiv, without language restrictions. We will include randomized controlled trials, quasi-randomized trials, and observational studies. Study selection and data extraction will be managed using Covidence. Two reviewers will independently screen titles and abstracts, assess full-text eligibility, evaluate risk of bias, and extract data.

**Data availability statement:** No datasets were generated or analysed during the current study. All relevant data from this study will be made available upon study completion.

**Funding:** The author(s) received no specific funding for this work.

**Competing interests:** The authors have declared that no competing interests exist.

## Ethics and dissemination

As this systematic review relies on the analysis of secondary data from published studies, ethical approval is not required. Findings will be disseminated through publication in a peer-reviewed journal and presented at relevant conferences.

## PROSPERO registration number

CRD42024617299

---

## Introduction

### Description of the condition

Limb amputation remains a significant global health burden, with updated estimates projecting that over 3.6 million individuals will be living with limb loss in the United States by 2050 [1]. The incidence continues to rise, driven largely by the aging population and the increasing prevalence of diabetes mellitus and peripheral vascular disease [1,2]. The most common etiologies for lower limb loss remain vascular pathology (primarily ischemia and diabetes), followed by trauma and malignancy. Conversely, upper extremity amputations in adults are predominantly traumatic in origin [2].

Demographic disparities persist as a critical issue; recent data from 2025 indicate that Black patients face up to a four-fold higher risk of major amputation compared to White patients, a gap often attributed to inequities in pre-amputation vascular care and socioeconomic determinants [3,4].

Beyond the physical loss, amputation is associated with significant morbidity, including high rates of surgical site infections, thromboembolism, and psychological distress such as anxiety and depression, which affect nearly 65% of patients in the early postoperative period [5]. Long-term mortality rates remain alarmingly high; recent cohorts report 1-year and 5-year mortality rates of approximately 24% and 48–60%, respectively, for major vascular amputations [6,7].

A critical yet often overlooked aspect of recovery is the development of neuropathic pain, primarily manifesting as Phantom Limb Pain (PLP) or Residual Limb Pain (RLP).

A central mechanism in the development of post-amputation pain is the formation of neuromas. Peripheral nerves are physiologically encased in an epineural sheath. During amputation, this sheath is disrupted, and severed axons regenerate in a disorganized manner into the surrounding scar tissue, forming a neuroma [8]. Although benign, these neuromas are frequently symptomatic, serving as ectopic foci of spontaneous neural discharge. The incidence of symptomatic neuromas varies significantly but has been reported to range from 4% to 49% in recent systematic reviews [9].

### Phantom Limb Pain and Residual Limb Pain

Phantom Limb Pain (PLP) is defined as the perception of painful sensations in the absent portion of the limb. It is a highly prevalent sequela, with recent narrative

reviews estimating a prevalence of over 60% in the chronic phase [10]. Its pathophysiology involves both peripheral mechanisms (neuroma discharge) and central mechanisms (cortical reorganization and spinal sensitization). PLP incidence appears largely independent of age, sex, or amputation level [10,11]. Furthermore, conservative treatments such as pharmacological management, physical therapy, and mirror therapy have yielded inconsistent preventive results [11,12].

Residual Limb Pain (RLP), distinct from PLP, is localized to the remaining stump and is often driven by the presence of symptomatic neuromas. RLP can prohibit the comfortable use of prosthetics and significantly impair quality of life. The psychological toll is profound, with patients frequently experiencing anxiety, post-traumatic stress, and affective distress [5,9].

### Description of the interventions

Recently, surgical strategies have shifted from simple nerve transection to prophylactic reconstruction to mitigate these pathological processes.

- Targeted Muscle Reinnervation (TMR): Involves transferring severed motor nerve endings to the motor branches of nearby target muscles, providing a physiological "target" for regenerating axons [13].

- Regenerative Peripheral Nerve Interface (RPNI): Involves wrapping the dissected nerve end in an autologous free muscle graft. The graft revascularizes and serves as a "bio-amplifier" and target for the nerve axons to innervate, similarly preventing neuroma formation by satisfying the biological drive for axonal regeneration [14,15].

Previous systematic reviews, such as those by De Lange et al. and Mauch et al., have demonstrated that prophylactic intervention with TMR or RPNI significantly reduces the incidence of PLP and RLP compared to classical amputation [16,17]. However, these reviews encountered a scarcity of comparative studies. To date, TMR and RPNI have seldom been compared directly to one another in a comprehensive manner. Furthermore, their comparative effectiveness across different patient demographics—including the racial and ethnic disparities mentioned above—and comorbidities (diabetes, vascular disease) remains unclear. This systematic review aims to fill this gap by providing a comparison of the safety and efficacy of TMR versus RPNI.

## Methods

### Protocol design and registration

This systematic review protocol was developed in accordance with the Preferred Reporting Items for Systematic Review and Meta-Analysis Protocols (PRISMA-P) statement [18,19]. The review will be conducted following the methodological guidance from the Cochrane Handbook for Systematic Reviews of Interventions and the Methodological Expectations for Cochrane Intervention Reviews (MECIR) standards [20]. The final report will comply with the PRISMA 2020 statement [21,22]. The protocol is registered in PROSPERO (CRD42024617299).

### Eligibility criteria

**Types of studies.** We will include studies written in any language, regardless of publication date. To ensure a comprehensive synthesis of the available evidence, we will include randomized controlled trials (RCTs) and non-randomized studies of interventions (NRSI), including quasi-randomized studies, cohort studies (prospective and retrospective), case-control studies, and case series with a sample size of n > 10.

We will exclude case reports, case series with n < 10, qualitative studies, and narrative reviews.

**Participants.** We will include adults (aged >18 years) of any gender who have undergone major upper or lower limb amputation of any etiology (e.g., trauma, vascular disease, malignancy, infection).

## Interventions and comparators

We will include studies comparing TMR or RPNI with classical amputation, as well as direct comparisons between TMR and RPNI. Additionally, comparisons of prophylactic versus curative applications of these techniques will be included for subgroup analysis.

For this review, "classical amputation" is defined as standard amputation (myoplasty or myodesy) where nerves are managed via traction neurectomy, simple ligation, or burial in muscle, without any additional specific reinnervation procedure or regenerative interface.

## Outcomes

Currently, no standardized Core Outcome Set (COS) exists for effectiveness trials regarding post-amputation neural reinnervation techniques. Consequently, the outcome measures for this review were selected based on a preliminary analysis of key systematic reviews and primary observational studies—specifically those by De Lange et al., Mauch et al., Yuan et al. and Zimbulis et al. [13,16,17,23]—to ensure clinical relevance and comparability with existing literature.

To determine both clinical benefit and risk, outcomes are categorized into efficacy and safety domains.

Primary efficacy outcomes include the incidence of symptomatic neuroma, specifically recording whether the diagnosis relied on standardized definitions such as Arnold's criteria [24]. These criteria classify a neuroma as symptomatic based on the presence of pain with specific characteristics in a defined neural distribution (mandatory), alongside confirmatory findings such as a positive Tinel sign, response to local anesthetic block, or imaging verification via ultrasound or magnetic resonance image. Additionally, we will assess the incidence and severity of both phantom limb pain (PLP) and residual limb pain (RLP), measured using validated scales such as the Visual Analog Scale (VAS), Numeric Rating Scale (NRS), or PROMIS.

Secondary safety outcomes encompass mortality and surgical complications, including wound dehiscence, surgical site infection (superficial or deep), hematoma, seroma, and abscess formation. We will also evaluate reoperation rates, defined as the need for stump revision, neuroma excision, or other secondary surgical procedures.

## Search methods for identification of studies

A comprehensive electronic search will be performed in the following databases from inception to the present, with no language restrictions: Cochrane Central Register of Controlled Trials (CENTRAL), Web of Science, Scopus, PubMed/MEDLINE, CINAHL, Google Scholar and MedRxiv (to identify unpublished/gray literature and minimize publication bias).

The full search strategy, including keywords and boolean operators, is provided in S1 File (search strategy).

## Data collection and analysis

**Selection of studies.** We will use Covidence software [25] for title and abstract screening. Two review authors (JM, MR) will independently screen all identified records. Full-text articles of potentially relevant studies will be retrieved and assessed independently by the same two authors against the eligibility criteria. Any disagreements regarding inclusion will be resolved through discussion or consultation with a third author (DG). A PRISMA flow diagram will document the selection process.

**Data extraction and management.** Two review authors will independently extract data using a piloted data extraction form. Disagreements will be resolved by consensus or with the third author (DG). We will extract the following data:

• Administrative: study author(s), year of publication, country, funding, and conflicts of interest.

• Methods: study design, sample size, inclusion/exclusion criteria, follow-up duration.

- Participants: number, age, sex, ethnicity, comorbidities (specifically diabetes, coronary heart disease, peripheral arterial disease), cause of amputation, and level of amputation.

- Intervention details: technique type (TMR/RPNI), level of amputation, previous nerve division into fascicles, and surgical timing.

- Confounders: surgical team specialty, adjuvant oncological treatment (neurotoxic chemotherapy or radiotherapy), and pre-operative pain status (to account for central sensitization).

- Outcomes: data on neuroma incidence, PLP/RLP (incidence and scores), and all listed safety complications.

**Assessment of risk of bias in included studies.** Two authors will independently assess the risk of bias using tools appropriate for each study design: the Cochrane Risk of Bias 2 (RoB 2) tool [26] for randomized trials, the Risk Of Bias In Non-randomized Studies of Interventions (ROBINS-I) tool [27] for non-randomized studies, and The Joanna Briggs Institute (JBI) Checklist for case series [28].

**Data synthesis and measures of treatment effect.** Statistical analysis will be performed using STATA 16 (StataCorp, TX, USA) [29].

Dichotomous results will be pooled and reported as odds ratios (OR) with 95% confidence intervals (CI). For continuous outcomes, we will use the mean difference (MD) with 95% CI when studies use the same measurement scale. However, as we anticipate variability in pain assessment tools (e.g., VAS vs. PROMIS), we will utilize the Standardized Mean Difference (SMD) to pool results across studies using different continuous scales.

**Dealing with missing data.** In the event of missing or unclear data, we will attempt to contact the corresponding authors of the primary studies via email to request the necessary information. If the data remain unavailable, we will attempt to extract values from graphical representations (e.g., figures or plots) using digital extraction tools. If data cannot be reasonably recovered, the study will be excluded from the specific quantitative synthesis (meta-analysis) but will be retained for the narrative synthesis.

**Assessment of heterogeneity.** Heterogeneity will be assessed visually via forest plots and statistically using the $I^2$ statistic [30]. An $I^2$ value >50% will indicate substantial heterogeneity. Due to the expected clinical variability (different surgeons, amputation levels, etiologies), we will use a random-effects model (DerSimonian-Laird method) for all meta-analyses [31].

**Subgroup analysis and investigation of heterogeneity.** If sufficient data are available, we will conduct subgroup analyses based on demographic characteristics (sex and ethnicity), cause of amputation (trauma versus vascular), type and level of amputation, presence of comorbidities (such as diabetes and vascular disease), and specific nerve handling techniques (e.g., previous division into fascicles).

A cumulative meta-analysis will also be performed based on the year in which each study was conducted.

**Meta-regression.** To explore the influence of continuous variables, specifically the age of participants, a meta-regression will be performed [32].

**Assessment of reporting biases.** We will conduct a comprehensive search for unpublished studies to minimize publication bias. If 10 or more studies are included in a meta-analysis, we will assess reporting bias graphically using a funnel plot and statistically using Egger's test. We will perform a Trim and Fill analysis to further assess the impact of potential publication bias.

**Sensitivity analysis.** We will conduct sensitivity analyses to assess the robustness of findings by excluding studies with a high risk of bias (defined as high risk in two or more domains) and by excluding or separately analyzing case series to explore the impact of study design. Furthermore, to enhance sample size validity, we will compare prophylactic TMR/RPNI arms from comparative studies with the corresponding prophylactic arms from mixed studies.

**Summary of findings**

We will use the GRADE approach (Grading of Recommendations Assessment, Development and Evaluation) to assess the certainty of the evidence for the primary outcomes [33].

**Ethics and dissemination**

No ethical approval is needed for this review, as we will use data from previously published studies. We will spread our research in national and international medical congresses and any other non-conflicting means of publication.

## Discussion

### Strengths and limitations of this study

**Strengths.**

- This constitutes the first systematic review to directly compare the clinical outcomes of Regenerative Peripheral Nerve Interface (RPNI) versus Targeted Muscle Reinnervation (TMR).

- Through a rigorous selection process compliant with PRISMA standards, this review will synthesize the totality of available evidence—ranging from randomized trials to observational studies—to evaluate the safety and efficacy of RPNI and TMR, assessing the certainty of findings using GRADE.

- To minimize potential publication bias, the search strategy extends to gray literature and unpublished studies sourced from MedRxiv.

**Limitations.**

- We anticipate a certain degree of heterogeneity among the included studies due to potential variations in surgical techniques and patient populations.

- The inclusion of case series may introduce susceptibility to bias inherent to non-comparative study designs; however, incorporating these studies is essential to provide a comprehensive overview of the current evidence landscape.

**Project timeline.** The review process has commenced. Pilot work and formal searching have been completed. Data extraction and risk of bias assessment are currently in progress and are expected to be completed by February 2026. Data synthesis and the final report are anticipated to be finalized by April 2026.

## Supporting information

**S1 File. Search strategy.** Full electronic search strategy, including keywords and boolean operators, for Cochrane Central Register of Controlled Trials (CENTRAL), Web of Science, Scopus, PubMed/MEDLINE, CINAHL and Google Scholar. (DOCX)

**S2 File. PRISMA checklist.** PRISMA-P 2015 checklist outlining where each item is reported within the manuscript. (DOCX)

## Acknowledgments

The authors would like to thank Dr. Miguel Ruiz Canela (Professor of Preventive Medicine and Public Health, University of Navarra) for his assistance in advising us in the methods section.

## Author contributions

**Conceptualization:** David Gurpegui.

**Data curation:** Jesús del Moral, David Gurpegui, Montserrat Royo.

**Formal analysis:** David Gurpegui.

**Investigation:** Jesús del Moral, David Gurpegui, Montserrat Royo.

**Methodology:** Jesús del Moral, David Gurpegui, Montserrat Royo.

**Project administration:** David Gurpegui.

**Resources:** Montserrat Royo.

**Software:** Montserrat Royo.

**Supervision:** David Gurpegui, Bernardo Hontanilla.

**Validation:** David Gurpegui, Bernardo Hontanilla.

**Writing – original draft:** Jesús del Moral.

**Writing – review & editing:** Jesús del Moral, David Gurpegui.

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
