## [Decision Letter · Decision Letter 0]

26 Nov 2025

PONE-D-25-31923Targeted Muscle Reinnervation (TMR) or Regenerative Peripheral Nerve Interface (RPNI) for pain prevention in patients with limb amputation: a protocol for a systematic review and meta-analysisPLOS ONE

Dear Dr. Gurpegui,

Thank you for submitting your manuscript to PLOS ONE. After careful consideration, we feel that it has merit but does not fully meet PLOS ONE’s publication criteria as it currently stands. Therefore, we invite you to submit a revised version of the manuscript that addresses the points raised during the review process.

A rebuttal letter that responds to each point raised by the academic editor and reviewer(s). You should upload this letter as a separate file labeled ’Response to Reviewers’.A marked-up copy of your manuscript that highlights changes made to the original version. You should upload this as a separate file labeled ’Revised Manuscript with Track Changes’.An unmarked version of your revised paper without tracked changes. You should upload this as a separate file labeled ’Manuscript’.

We look forward to receiving your revised manuscript.

Kind regards,

Simone Ranaldi, Ph.D.

Academic Editor

PLOS ONE

Journal Requirements:

1. Please ensure that your manuscript meets PLOS ONE’s style requirements, including those for file naming. The PLOS ONE style templates can be found at

**Additional Editor Comments:**

The two independent reviewers identified issues with the level of detail that is included in the manuscript, the list of reference and the general grammar and syntax. All of these issues should be solved fully before publication in the Journal.

Reviewers’ comments:

Reviewer’s Responses to Questions

**Comments to the Author**

1. Does the manuscript provide a valid rationale for the proposed study, with clearly identified and justified research questions?

Reviewer #1: Yes

Reviewer #2: Partly

2. Is the protocol technically sound and planned in a manner that will lead to a meaningful outcome and allow testing the stated hypotheses?

Reviewer #1: Yes

Reviewer #2: Partly

3. Is the methodology feasible and described in sufficient detail to allow the work to be replicable?

Reviewer #1: Yes

Reviewer #2: No

4. Have the authors described where all data underlying the findings will be made available when the study is complete?

Reviewer #1: Yes

Reviewer #2: Yes

5. Is the manuscript presented in an intelligible fashion and written in standard English?

Reviewer #1: Yes

Reviewer #2: No

6. Review Comments to the Author

You may also provide optional suggestions and comments to authors that they might find helpful in planning their study.

Reviewer #1: Summary of Key Recommendations

Area Recommendation

Method clarity Define “classical amputation” more precisely

Data synthesis Clarify standardization for different pain scales

Limitations Add a dedicated limitations section

Figures Include a PRISMA-P diagram

Supplementary Include the full search strategy as a supplement

Language Minor editorial corrections for grammar and clarity

Reviewer #2: Extent of my review

My review put the overall content of this manuscript in perspective with previous works involving qualitative and quantitative assessment of clinical outcomes.

I have also focused mainly on the strength of the argumentation, the validity of the study protocol and overall coherence of the manuscript.

Relevance of this work

The authors aim to address genuine and important clinical questions: what are the true clinical benefits of surgical reinnervations and what are difference of outcomes between TMR and RNPI?

The study protocol including systematic review and meta-analysis is conventional.

Clearly, there is a need for this type of publication that will be critical to all stakeholders, including consumers, practitioners and decision makers.

Limitations of this work

My overall impression is that the manuscript is written conservatively providing only the minimal required information. Consequently, it does fail short of providing comprehensive description and, sometime, critical elements of the study protocol.

For example:

• Outcome measures are not categorized according to safety and efficacy,

• Examples of searches syntaxes in online supplemental material 1 not available with the submission,

• The data extraction form is missing,

• The table for the timeline is not referenced,

• The heading for “references” is missing,

• Nearly half of the references are more than 10-year-old,

Altogether, the manuscript will benefit from professional editing to improve the overall English writing style (e.g., wording and syntax).

Outcomes

Despite these concerns, I do believe that the core work of the study can be original and worthwhile publishing provided that the authors address the points below. I am confident that a substantial publication that will be well-cited could come out of the work.

Selected specific comments

1. L 25: Consider adding the lower time limit for the reference selection, if any.

2. L 24: Re-word “what each technique offers” in terms of efficacy and safety of RPNI and TMR.

3. L 38: Verify if RPNI and TMR are acknowledged MeSH terms (e.g., need to spell them out).

4. L 43: Modify this statement about “high-quality articles” given the proposed selection criteria including not only randomized and quasi-randomized with high level of evidence, but also observational studies which could potentially include case series with low level of evidence. The next point is also referring to including gray literature with low level of evidence

5. L 49: Replace “superior or inferior limb amputation” by more commonly used “upper or lower limb amputation”.

6. L 54: Add “Classical amputation without reinnervation”.

7. L 56: Categorize outcomes according to measure of efficacy and safety to aligned with the objectives in L 158.

8. L 71: Rewrite the sentence as “being age, renal disease, proximal amputation, diabetes, and peripheral vascular disease the risk factors” is unclear.

9. L 77-80: Add some statistical data to support these points.

10. L 82: Verify if “(the process of)” should be between brackets.

11. L 90: Verify if the sentence is completed.

12. L 92: Table 1 should include the spelling of US and MRI

13. L 95: Describe what is the “100%” in these stats (e.g., percentage of what).

14. L 111: Include information about other treatments for PLP such as mirror therapies.

15. L 123: Provide references looking at one or the other or both interventions.

16. L 165: Included other critical confounders such as time since amputation, time since reinnervation, daily usage of prosthesis.

17. L 200: Indicate clearly to outcomes related to efficacy and safety.

18. L 212: Provide list of keywords used to conduct the searches and examples of searches syntaxes (Online supplemental material 1 not available with the submission).

19. L 226: Reference the Table for timeline

20. L 239: Provide the draft of the data extraction form.

21. L 266: Spell IPC in caption of Table

22. L 314: Add heading for “References”,

23. L 315: Verify if any of 41 references more than 10-year old can be replaced by newer references.

7. PLOS authors have the option to publish the peer review history of their article (what does this mean?). If published, this will include your full peer review and any attached files.

Reviewer #1: **Yes:** Vaikunthan Rajaratnam

Reviewer #2: No

---

## [Author Response · Author response to Decision Letter 1]

10 Jan 2026

Response to Reviewer #1

General Comment: “Does the manuscript provide a valid rationale...? Yes. Is the protocol technically sound...? Yes.”

Response: We thank the reviewer for their positive assessment of our rationale and technical soundness.

1. Method clarity: Define “classical amputation” more precisely.

Response: We agree that this term can be ambiguous. We have added a precise definition in the Methods section. We now explicitly state that "classical amputation" refers to standard amputation techniques (myoplasty or myodesy) performed without any additional prophylactic nerve handling procedures such as TMR or RPNI.

2. Data synthesis: Clarify standardization for different pain scales.

Response: We have updated the Data Synthesis section to clarify that when studies use different continuous scales to measure the same outcome (e.g., VAS 0-10 vs. PROMIS measures), we will use the Standardized Mean Difference (SMD) rather than the Mean Difference (MD) to pool the results in the meta-analysis. This ensures statistical comparability.

3. Limitations: Add a dedicated limitations section.

Response: We have added a dedicated "Strengths and Limitations" section to the manuscript (previously part of the discussion/intro) to explicitly discuss potential limitations, such as the expected heterogeneity of the included studies and the potential inclusion of case series.

4. Figures: Include a PRISMA-P diagram.

Response: We have included the PRISMA-P checklist and ensuring the flow of the protocol aligns with PRISMA-P guidelines. A PRISMA flow diagram will be generated once the review is completed (results stage), but we have referenced the PRISMA-P checklist in the supplementary files as requested.

5. Supplementary: Include the full search strategy as a supplement.

Response: We apologize if this file was not accessible in the previous submission. We have ensured that the full search strategy (listing keywords and syntax for all databases) is correctly uploaded and labeled as S1 File.

6. Language: Minor editorial corrections for grammar and clarity.

Response: We have conducted a comprehensive linguistic revision of the entire manuscript. In addition to addressing specific grammatical errors, we have extensively edited the text to enhance its clarity, flow, and readability, consistent with the feedback provided by Reviewer #2.

Response to Reviewer #2

General Comment: “The core work of the study can be original and worthwhile publishing provided that the authors address the points below.”

Response: We sincerely thank the reviewer for seeing the potential in our work and for the detailed critique which has helped us refine the protocol significantly.

Critique on Style

We sincerely appreciate the reviewer’s observation regarding the writing style and the need for professional editing. We have taken this feedback very seriously. Furthermore, we have conducted a comprehensive linguistic revision and rewriting of the entire manuscript to improve clarity, flow, and academic tone. Consequently, the Introduction and Methods sections have been substantially rephrased to meet the journal’s standards.

This extensive revision addresses the specific stylistic points raised:

L 49 (Point 5): Terminology has been standardized to “upper or lower limb amputation” throughout the text.

L 24, 71, 82, 90, 95 (Points 2, 8, 10, 11, 13): These sentences have been completely rewritten or integrated into new paragraphs to ensure grammatical correctness and better readability.

L 266 (Point 21): Spelling errors (e.g., IPC) have been corrected or eliminated.

Critique on References (Old references) & Documentation:

Response: We agree with this observation. We have updated the bibliography to prioritize recent literature (2020–2025), particularly in the Introduction regarding amputation epidemiology and pain management, to ensure the clinical context is up to date. Additionally, we have verified that all supplementary materials (Search Strategy, Data Extraction Form) are correctly attached and accessible.

Response to Specific Technical Comments:

1. L 25: Consider adding the lower time limit for the reference selection, if any.

Response: We have clarified in the Methods that there are no lower time limits for study inclusion to ensure comprehensive coverage. Anyway, we expect most relevant TMR/RPNI studies to be recent.

3. L 38: Verify if RPNI and TMR are acknowledged MeSH terms.

Response: We have verified this. While TMR and RPNI are not yet specific MeSH headings, we agree that spelling them out improves discoverability. We have updated the Keywords section to include the full terms "Targeted Muscle Reinnervation" and "Regenerative Peripheral Nerve Interface".

4. L 43: Modify this statement about “high-quality articles”…

Response: We agree that the original phrasing was contradictory given our inclusion criteria. We have revised the “Strengths and Limitations” section to state: “-Through a rigorous selection process compliant with PRISMA standards, this review will collate the totality of the available evidence to evaluate the safety and efficacy of RPNI and TMR, assessing the certainty of findings using GRADE.”

6. L 54: Add “Classical amputation without reinnervation”.

Response: Added for clarity.

7. L 56 & L 200: Categorize outcomes according to measure of efficacy and safety.

Response: We have reorganized the Outcomes section. We now explicitly categorize them under “Efficacy Outcomes” (Neuroma, PLP, RLP, Pain scores) and “Safety Outcomes” (Surgical complications).

9. L 77-80: Add some statistical data to support these points.

Response: We have extensively revised this section in the Introduction. We now explicitly cite recent projections estimating over 3.6 million individuals living with limb loss by 2050, as well as specific data on demographic disparities, such as the four-fold higher risk of major amputation in Black patients [Refs 1-4].

12. L 92: Table 1 should include the spelling of US and MRI.

Response: To improve the flow of the manuscript, we have integrated the information regarding diagnostic criteria for neuromas directly into the Methods section. We have clarified that we will extract data based on study definitions, noting when standardized criteria like Arnold’s are used. Within that text, we have ensured that the abbreviations are fully spelled out as “Ultrasound” and “Magnetic Resonance Imaging”.

14. L 111: Include information about other treatments for PLP.

Response: We have briefly mentioned other treatments (e.g., mirror therapy, medication) to provide context on why surgical prophylaxis is distinct/important.

15. L 123: Provide references looking at one or the other or both interventions.

Response: We have ensured that all statements regarding the individual efficacy of TMR and RPNI are now supported by specific, recent references. In the Introduction, we explicitly cite Zimbulis et al. [12] for TMR efficacy, Hooper et al. [13] and Kubiak et al. [14] for RPNI outcomes, and De Lange et al. [15] and Mauch et al. [16] as systematic reviews covering both interventions.

16. L 165: Included other critical confounders.

Response: Excellent suggestion. We have added “Surgical team specialty”, “adjuvant oncological treatment (neurotoxic chemotherapy or radiotherapy)” and “pre-operative pain status (to account for central sensitization)” to our list of data to be extracted and considered in subgroup analyses or meta-regression, as these are critical effect modifiers.

17. L 200: Indicate clearly to outcomes related to efficacy and safety.

Response: Please refer to our response to Comment #7. We have applied this categorization throughout the manuscript, clearly distinguishing between Efficacy and Safety outcomes in the Methods section.

18. L 212 & 20 (L 239): Provide list of keywords and data extraction form.

Response: The complete Search Strategy (keywords and syntax) has been uploaded as Supplementary Material (S1 File). Regarding the Data Extraction Form, we utilize Covidence software for the review process; therefore, the form is a digital template integrated within the platform rather than a standalone document. However, to ensure full transparency and comply with your request, we have explicitly listed every variable and field used in this digital template within the "Data Extraction and Management" section of the manuscript.

19. L 226: Reference the Table for timeline.

Response: To streamline the manuscript, we have replaced the timeline table with a narrative description. The project schedule and updated completion dates (2026) are now explicitly detailed in the text of the Methods section (under the subheading Project Timeline).

21. L 266: Spell IPC in caption of Table

Response: The table containing this abbreviation has been removed during the manuscript revision to streamline the text. The relevant information is now presented in narrative form in the Methods section, ensuring no undefined abbreviations remain.

22 & 23 (L 314-315): Heading for “References” and older references.

Response: We have added the “References” heading. Furthermore, we have conducted a major update of the bibliography, replacing older statistics with literature from 2015-2025 to ensure the protocol reflects the current state of the art.

We hope that the revised manuscript now meets the high standards of PLOS ONE. We look forward to your decision.

---

## [Decision Letter · Decision Letter 1]

19 May 2026

Targeted Muscle Reinnervation (TMR) or Regenerative Peripheral Nerve Interface (RPNI) for pain prevention in patients with limb amputation: a protocol for a systematic review and meta-analysis

PONE-D-25-31923R1

Dear Dr. Gurpegui,

We’re pleased to inform you that your manuscript has been judged scientifically suitable for publication and will be formally accepted for publication once it meets all outstanding technical requirements.

An invoice will be generated when your article is formally accepted. Please note, if your institution has a publishing partnership with PLOS and your article meets the relevant criteria, all or part of your publication costs will be covered. Please make sure your user information is up-to-date by logging into Editorial Manager at Editorial Manager® and clicking the ‘Update My Information’ link at the top of the page. For questions related to billing, please contact billing support.

Kind regards,

Yee Gary Ang, MBBS MPH

Academic Editor

PLOS One

Additional Editor Comments (optional):

Reviewers’ comments:

Reviewer’s Responses to Questions

**Comments to the Author**

1. Does the manuscript provide a valid rationale for the proposed study, with clearly identified and justified research questions?

Reviewer #2: Yes

2. Is the protocol technically sound and planned in a manner that will lead to a meaningful outcome and allow testing the stated hypotheses?

Reviewer #2: Yes

3. Is the methodology feasible and described in sufficient detail to allow the work to be replicable?

Reviewer #2: Yes

4. Have the authors described where all data underlying the findings will be made available when the study is complete?

Reviewer #2: Yes

5. Is the manuscript presented in an intelligible fashion and written in standard English?

Reviewer #2: No

6. Review Comments to the Author

You may also provide optional suggestions and comments to authors that they might find helpful in planning their study.

Reviewer #2: Overall comments

General Comments

My overall impression is that the authors have carefully read the comments and genuinely made significant efforts to address all comments thoroughly both in the manuscript and the supplement material. The authors addressed satisfactorily all my initial comments. I have made a few new suggestions to address some minor points in the revised manuscript.

Outcomes

I believe that the core of this study is original and worthwhile publishing. In my view, the manuscript is ready for publication upon consideration of the comments below.

Selected specific comments

Introduction

1.L 46: Consider replacing “patients” by more accepted term such as “individuals”, “users” or “participants” throughout the manuscript depending on the context.

2.L 45: Make sure the term “Black” and “White” participants are correct. Consider using “Caucasian” rather than “White”

3.L 48: Verify if the term “morbidity” should be replaced by “comorbidity”. This will make more sense to me.

4.L 72: Replacing “stump” by more accepted clinical term “residuum” throughout the manuscript

Methods

5.L 112: Consider naming “case series” for studies with n<10 and “cohort studies” for studies with N>10. This will simply the description of the limitations.

6.L 118: Consider adding a sentence about the inclusion of all ethnic background to clearly show the link with the objective about Black and Caucasian backgrounds.

7.L 133: Correct “benefit and risk” by “benefits and risks”.

8.L 142: Spell out “Patient-Reported Outcomes Measurement Information System (PROMIS)”.

9.L 209: Clarify if the analysis should also be stratified by ethnic background.

Discussion

10.L 247: Rephrase the sentence to avoid confusion with the study excluded. See previous comments about “cohort studies” vs “case series”.

References

11.L 200: Provide publication details for Ref 13.

7. PLOS authors have the option to publish the peer review history of their article (what does this mean?). If published, this will include your full peer review and any attached files.

Reviewer #2: No

---

## [Editor Report · Acceptance letter]

PONE-D-25-31923R1

PLOS One

Dear Dr. Gurpegui,

I’m pleased to inform you that your manuscript has been deemed suitable for publication in PLOS One. Congratulations! Your manuscript is now being handed over to our production team.

Lastly, if your institution or institutions have a press office, please let them know about your upcoming paper now to help maximize its impact. If they’ll be preparing press materials, please inform our press team within the next 48 hours. Your manuscript will remain under strict press embargo until 2 pm Eastern Time on the date of publication. For more information, please contact onepress@plos.org.

Kind regards,

on behalf of

Dr. Yee Gary Ang

Academic Editor

PLOS One